# EGF and IgA in maternal milk, donor milk, and milk fortifiers in the neonatal intensive care unit setting

**Christian Tamar**[1], **Kara Greenfield**[1], **Katya McDonald**[1], **Emily Levy**[2], **Jane E. Brumbaugh**[2], **Kathryn Knoop**[1,2]*

1 Department of Immunology, Mayo Clinic, Rochester, Minnesota, United States of America, 2 Department of Pediatric and Adolescent Medicine, Mayo Clinic, Rochester, Minnesota, United States of America

* knoop.kathryn@mayo.edu

## Abstract

Human milk contains a variety of factors that positively contribute to neonatal health, including epidermal growth factor (EGF) and immunoglobulin A (IgA). When maternal milk cannot be the primary diet, maternal milk alternatives like donor human milk or formula can be provided. Donor human milk is increasingly provided to infants born preterm or low birth weight with the aim to supply immunological factors at similar concentrations to maternal milk. We sought to assess the concentrations of human EGF and IgA in the diet and stool of neonates between exclusive maternal milk, donor human milk, or formula-based diets. Using a prospective cohort study, we collected samples of diet and stool weekly from premature and low birth weight neonates starting at 10 days post-natal through five weeks of life while admitted to a neonatal intensive care unit (NICU). Compared to formula, there was significantly more EGF in both the milk and the stool of the infants fed human milk. Donor milk pooled from multiple donors contained similar concentrations of EGF and IgA to maternal milk, which was also significantly more than formula diets. Maternal milk supplemented with a fortifier derived from human milk contained significantly more EGF and IgA compared to unfortified maternal milk or maternal milk supplemented with fortifier derived from bovine milk. Further analysis of human milk-derived fortifiers confirmed these fortifiers contained significant concentrations of EGF and IgA, contributing to an increased concentration of those factors that bovine milk-derived fortifiers do not confer. These findings illustrate how the choice of diet for a newborn, and even how that diet is modified through fortifiers or pasteurization before ingestion, impacts the beneficial biomolecules the infant receives from feeding.

## Introduction

Every year approximately 10% of infants in the United States are born prematurely at less than 37 weeks gestational age, putting the infant at risk for a myriad of developmental and chronic diseases [1]. Of prematurely born infants, those born at very low birth weight (VLBW), defined as less than 1500 grams [2], represent around 1% of all live births in the US [1]. These VLBW infants often spend weeks in the neonatal intensive care unit (NICU) as they develop

**Data availability statement:** All relevant data are within the paper and its Supporting Information files. The minimal data set in the supporting file includes the data used to generate the graphs for the figures.

**Funding:** KK NIH DK134366.

**Competing interests:** The authors have declared that no competing interests exist.

and stabilize outside of the uterus. During their stay in the NICU, VLBW infants are highly vulnerable to opportunistic and nosocomial pathogens. Maternal milk, defined here as milk received from the infant's biological mother, is recommended as the exclusive diet for infants during the critical first six months of life by the American Academy of Pediatrics and World Health Organization [3]. Early enteral feeding with maternal milk reduces incidence of, and mortality from, infectious diseases, including outcomes like neonatal late-onset sepsis [4] and necrotizing enterocolitis [5, 6].

Many factors in human milk confer neonatal immunity and temper inflammatory responses to unfamiliar antigens, protecting against infection and shaping the infant microbiome [7–9]. One of these human milk factors, epidermal growth factor (EGF), has been shown to improve intestinal barrier function by promoting epithelial cell growth and decreasing bacterial translocation through the intestinal epithelium to the bloodstream in maternal milk-fed infants [10–13]. Furthermore, in a mouse model of neonatal sepsis, EGF improved intestinal barrier function in neonates, prevented enteric pathogens translocating from the intestine, and prevented the development of sepsis secondary to bloodstream infection [14]. While EGF directly strengthens the intestinal epithelium, the human milk factor immunoglobulin A (IgA) contributes to neonatal intestinal health primarily through its interactions with the developing microbiome. In addition to encouraging the growth of bacterial commensals, maternal IgA provides passive immunity to the neonate through targeted protection against immunologically relevant antigens before the infant gains the ability to generate their own IgA response, reducing the risk of enteric infection and necrotizing enterocolitis [6,15–17].

When the maternal milk containing these valuable factors is unavailable, Donor human milk and human milk-derived fortifiers are increasingly utilized for VLBW infants rather than formula [18, 19]. Donor milk can be expressed throughout a donor's lactation cycle, often months following parturition when the concentrations of many milk factors, including EGF and IgA, are lower than immediately after birth [20–23]. We have observed in our preclinical murine model of neonatal sepsis that milk expressed closer to parturition contained more EGF and offered more protection from enteric pathogens than milk expressed later in lactation [14]. This gradual decline throughout lactation was also reflected in the stool of human infants fed maternal milk but not in formula-fed infants [14]. Donor milk is often provided in this asynchronous manner, with the timing of donor milk collection during lactation unlikely to be matched to the infant's corrected gestational age. To address how much the composition of human milk can naturally vary between donors and minimize the impact of individual sample differences, some milk banks pool their donor milk from multiple donors [24].

In addition to the choice of primary diet between maternal milk, donor milk, and formula, a further measure to address the high nutritional demands of preterm or VLBW infants is the introduction of a nutritional supplement to their milk in the form of a fortifier. A goal of supplementing milk with fortifiers is to increase caloric and mineral content of the milk and is commonly practiced in NICUs because unfortified milk does not meet the nutritional needs of preterm infants [25]. Fortifiers can be derived from different origins, including human and bovine milk, and can vary substantially in their nutritional composition, with potential ramifications for their immunological benefit [25]. Fortifiers are designed to be added directly to human milk, not infant formula, at specified dilutions, and can have a frozen shelf-life of up to two years. As of 2018, over 90% of level 3 and 4 NICU facilities reported using fortifiers, particularly in VLBW infant populations [26].

In this study, we analyzed the concentrations of EGF and IgA in maternal milk, donor milk, and formula provided to neonates. In addition to these diets, we measured EGF in the stool of infants to determine if the amount of EGF present in the stool reflected the amount received in the diet. Finally, we examined EGF and IgA concentrations in different types of

human milk fortifiers. Here, we show that EGF and IgA concentrations vary between maternal milk and its commonly used alternatives, including between individual and pooled donor milk, and can be further altered by the addition of different types of human milk fortifiers, with potential functional consequences.

## Methods

### Human subjects

Mother-infant paired participants were recruited for this observational study between March 2021-March 2023 based on infant birth weight (<2500 grams) or birth at <36 weeks gestation. Infant participants were at least 3 days old at enrollment and admitted to the Mayo Clinic NICU, in Rochester, Minnesota, USA. Beginning at 8 days following birth or later, milk and infant stool were collected weekly for 5 weeks while hospitalized. Exclusion criteria included major congenital anomalies involving the intestinal tract and placement on *nil per os* (NPO) orders. Written informed consent was obtained from a parent of eligible infants. Clinical data including gestational age at birth, birth weight, birth length, frontal occipital circumference, mode of delivery, age at full feeding, history of culture positive sepsis, use of steroids or antibiotics, and diet including fortification was collected from infant participant medical charts and deidentified by study coordinator. Research technicians performing assays were blinded to participant identity. Samples were collected beginning at 8 days following delivery and therefore avoided measuring colostrum, which can contain distinct composition compared to mature milk [27]. As this was a prospective observational study, all eligible mother-infant pairs were approached, and consented pairs were included in the study during the two-year period, regardless of infant diet during the study. This study was approved through the Mayo Clinic Institutional Review Board.

### Fortifiers and formula

Some infant participants' diets were supplemented with either bovine milk-based fortifiers (Similac Human Milk Fortifier or Enfamil Human Milk Fortifier) or human milk-based fortifiers (Prolact+6 H²MF, Prolact+8 H²MF Human Milk Fortifier) at the discretion of the clinical care team. Such fortifiers are designed to be added to human milk, and are added at specified dilutions. Enfamil and Similac Human Milk Fortifiers were prepared according to manufacturers' instructions at either 1:6 or 1:11 dilution in milk. Prolacta products were also prepared according to preparation directions, with Prolact+6 diluted 3:10 or Prolact+8 diluted 2:5 in milk. In some experiments, fortifiers were used for experimental assays and aliquots of fortifiers were obtained for direct analysis from the Mayo Clinic Nutrition Lab. For direct analysis of fortifiers, independent lots were used as biological replicates. Some infant participants were provided infant formula as opposed to human milk. Formula used was either Enfamil Enfacare, 22 calories per ounce, or Similac Specialcare, 24 calories per ounce. Neither of these formulations have added epidermal growth factor or immunoglobulins, and therefore are not expected to contain human EGF or IgA.

### Milk and stool collection

Diet samples refer to infant diet, and consisted of expressed maternal milk, donor milk, or infant formula, as available or based on the discretion of the clinical care team. A 2 mL aliquot of the diet for the infant participant was saved from the prepared milk by the nursing staff immediately prior to bottle or gavage feeding and stored at 4°C until collection for processing for no more than 4 hours. Between 24 and 48 hours later, stool was collected from the

diaper by nursing staff using a sporked collection tube and stored at -20°C until collection for processing within 4 hours. Incomplete sample pairs were discarded, therefore if either stool or milk from an individual infant participant was missing the corresponding milk or stool in a weekly paired collection, that sample was discarded. Only paired samples of both diet and stool from an individual in a given week were collected were analyzed. Maternal milk, following expression, was stored at 4°C until utilized for feed. Pooled donor milk was obtained from the OhioHealth Mothers' Milk Bank, a member of HMBANA (Human Milk Banking Association of North America). The OhioHealth Mothers' Milk Bank, and the HMBANA promotes collection and distribution of donor human milk in a safe, ethical, and cost effective manner [28]. Provided donor milk was pasteurized, and pooled from 3–5 individuals at the OhioHealth Mothers' Milk Bank, and batch information was collected. In some cases, fortifiers were added to human milk (donor milk or maternal milk) to increase nutritional content of the human milk. Diet was fortified with either bovine-based fortifiers (Similac Human Milk Fortifier or Enfamil Human Milk Fortifier) or human milk-derived fortifiers (Prolacta +4, +6, +8) by the nutrition team at the direction of the clinical care team. Milk type and fortifiers were noted by staff at time of collection. To compare pooled donor milk to individual donor milk, individual donor milk was used for some experimental assays. Individual donor milk was obtained from the Minnesota Milk Bank for Babies. Pasteurized milk frozen at -80°C from 20 different individual donors was obtained and kept separate. Time of collection post-partum was noted.

## Sample preparation

For stool sample preparation, 100 mg of stool was taken from the sample and homogenized in 1 mL of phosphate buffered saline (PBS) by a Bullet Blender Storm 24 Homogenizer (Model BBY24M, Next Advance, Troy, NY, USA) using 0.5 mm diameter zirconium oxide beads (Next Advance, Troy, NY, USA).

## ELISA

Human milk samples, human milk fortifiers, and human stool samples were analyzed by enzyme-linked immunoabsorbent assay (ELISA) for human IgA (88–50600, Invitrogen, Waltham, MA, USA) and human EGF (DY236, R&D Systems, Minneapolis, MN, USA) according to the manufacturer's protocol. Milk samples analyzed for human IgA content were diluted from 1:1000–1:20,000, samples analyzed for human EGF content were diluted from 1:500–1:2500, and human milk fortifiers were diluted at 1:10. Samples were diluted with a 1:20 dilution of Assay Buffer A Concentrate (PBS with 1% Tween 20 and 10% BSA, Invitrogen 88-50600-88) in deionized water. Sample optical density was measured using a BioTek 800 TS absorbance reader (BioTek, Winooski, VT, USA) at 450nm and 570nm. Modified ELISAs to measure human IgA capable of binding bacterial species began by coating a 96-well ELISA plate with either heat-killed *Streptococcus agalactiae* (Group B Strep) or heat-killed *Escherichia coli* diluted in phosphate-buffered saline (PBS) to $4 \times 10^5$ CFUs per well. The plate was incubated overnight at 4°C, then subsequently adhered to the manufacturer's protocol (Invitrogen 88–50600) to detect human IgA.

## Statistics

All statistical analysis was done using GraphPad Prism 9.0 (GraphPad Software Inc., Boston, MA, USA). Data is reported as mean +/- the standard error of the mean. Differences in EGF concentrations, IgA concentrations, or absorbance units indicating binding ability of IgA to bacteria between groups were analyzed using the non-parametric Kruskal-Wallis test

when comparing three or more groups. To correct for multiple comparisons using statistical hypothesis testing, Dunn's post-hoc test for multiple comparisons was applied comparing the mean rank of every group to every other group. When comparing bovine milk-derived fortifier directly to human milk-derived fortifier, the non-parametric Mann-Whitney test was used to compare these two groups. To compare EGF in diet and corresponding stool, or to compare EGF and IgA in diet Pearson's Correlation was used to identify a potential linear relationship.

## Results

### EGF is decreased in individual donor milk, but not pooled donor milk

Our enrolled cohort consisted of 74 mother-infant dyads, with 237 total diet samples collected which were recorded as either maternal milk (218), donor milk (13), or formula (6). 243 stool samples were also collected from the same infants. 20 samples of pasteurized milk from individual donors were obtained from the Minnesota Milk Bank for Babies for further comparison. Samples were collected beginning at 8 days following delivery to avoid measuring colostrum, which can contain distinct composition compared to mature milk [27]. Additional clinical characteristics of the cohort are displayed in Table 1. Relative to maternal milk (59.72 ± 2.25 ng/mL), both formula (9.54 ± 1.48 ng/mL) and individual donor milk samples (23.74 ± 2.29 ng/mL) had significantly decreased EGF (p=0.0001, p<0.0001; Fig. 1A). Pooled donor milk samples (70.15 ± 7.14 ng/mL) contained similar EGF concentrations as compared to

**Table 1. Description of Clinical Cohort.**

| Mother-infant Pairs | 74 | Total Diet Samples | 237 |
|---|---|---|---|
| Sex – Male/Female | 35/39 | Maternal Milk Samples | 218 |
| Delivery – Vaginal/C-section | 19/55 | Donor Milk Samples | 13 |
| Gest. Age (Week) | 29.6 ± 0.3 | Formula Samples | 6 |
| Birth Weight (Grams) | 1373 ± 53 | Total Stool Samples | 237 |

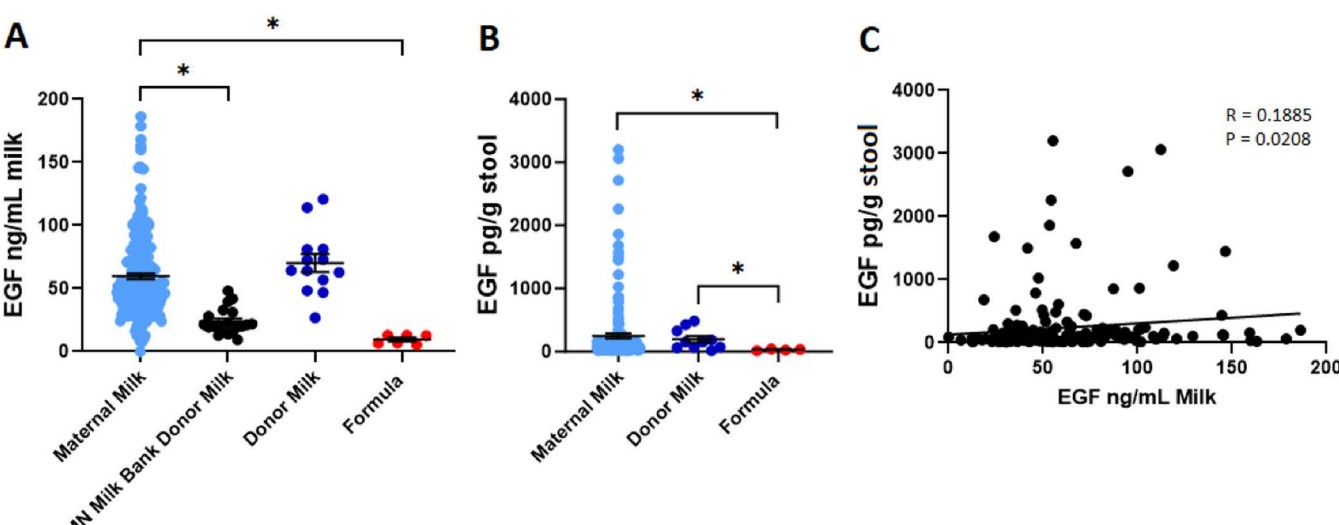

**Fig 1. EGF is decreased in individual donor milk, but not pooled donor milk.** A) Concentration of EGF in maternal milk (light blue, n=218), individual donor milk (black, n=20), pooled donor milk (dark blue, n=13), and formula (red, n=6). B) Concentration of EGF in stool from infants fed maternal milk (light blue, n=163), donor milk (dark blue, n=10), or formula (red, n=4). C) EGF in stool plotted against EGF in milk from matched-pairs specimens. * denotes significance, p<0.05, Kruskal-Wallis test with Dunn's multiple comparisons test in A and B, Pearson's correlation in C.

maternal milk samples (p=0.6630; Fig. 1A). To determine if the amount of EGF in the stool of infants reflects the initial amount of EGF from the diet, stool was collected from the infants receiving these milk or formula samples and evaluated for EGF content (Fig. 1B). Infants fed maternal milk (248.33 ± 40.62 pg/g) or pooled donor milk (194.59 ± 51.85 pg/g) had significantly more EGF in their stool on average than those given formula (29.06 ± 5.47 pg/g; p=0.0437, p=0.0193). We observed a weak positive correlation between infants' stool EGF and the EGF contents of paired milk or formula samples (Fig. 1C).

## Human milk-based fortifiers contribute to EGF concentration

We next assessed if fortifiers affected the concentration of EGF by comparing the fortified or unfortified milk (Fig. 2A). Fortified maternal milk (65.29 ± 2.86 ng/mL) had significantly increased EGF compared to unfortified maternal milk (49.11 ± 3.28 ng/mL; p=0.0061), though this was not the case in fortified donor milk as compared to unfortified donor milk (p>0.9999). The significant differences of EGF in the milk were not detected in infant stool (p>0.9999, p>0.9999; Fig. 2B). Following assessment based on type of fortifier used (bovine milk-derived or human milk-derived), we observed milk fortified with a human milk-derived fortifier contained significantly more EGF (74.16 ± 3.43 ng/mL) as compared to unfortified maternal milk (49.11 ± 3.31 ng/mL; p<0.0001) or milk fortified with a bovine milk-derived fortifier (41.29 ± 2.57 ng/mL; p<0.0001; Fig. 2C). Measuring fortifiers directly, only human milk-derived fortifier products exhibited a substantial level of human EGF (p=0.0070; Fig. 2D).

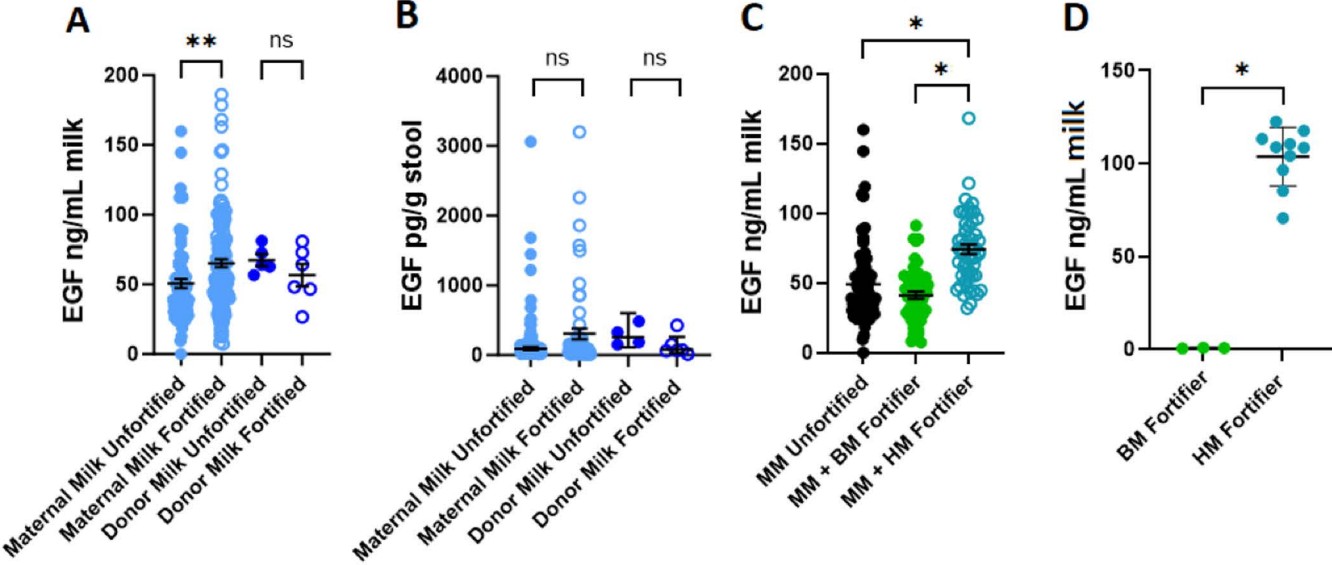

**Fig 2. Human milk-based fortifiers contribute to EGF concentration.** A) Concentration of EGF in maternal milk (light blue) or donor milk (dark blue), unfortified (solid circles) or fortified (open circles) (maternal milk unfortified: n=75; maternal milk fortified: n=143; donor milk unfortified: n=5; donor milk fortified: n=6). B) Concentration of EGF in stool from infants fed maternal milk (light blue) or donor milk (dark blue), unfortified (solid circles) or fortified (open circles) (maternal milk unfortified: n=90; maternal milk fortified: n=62; donor milk unfortified: n=4; donor milk fortified: n=6). C) Concentration of EGF in unfortified maternal milk (black, n=75), maternal milk fortified with bovine milk-based fortifier (green, n=55), or maternal milk fortified with human milk-based fortifier (blue, n=45). D) Concentration of EGF in bovine milk-based fortifier (green) and human milk-based fortifier (blue). * denotes significance, p<0.05 Kruskal-Wallis test with Dunn's multiple comparisons test in A, B, and C, and Mann-Whitney in D.

## IgA is decreased in individual donor milk and is present in human milk-based fortifiers

We then evaluated the potential variation of IgA in maternal and donor milk. As with EGF, maternal milk (346.35 ± 17.91 ug/mL) and pooled donor milk samples (327.86 ± 93.01 ug/mL) had substantially higher IgA concentrations compared to individual donor milk (93.64 ± 12.78 ug/mL; p<0.0001, p=0.0361) and formula samples (18.76 ± 4.05 ug/mL; p<0.0001, p=0.0197; Fig. 3A). We observed a weak positive correlation between the IgA concentrations of milk samples and those samples' corresponding EGF concentrations (Fig. 3B). We found increased IgA in maternal milk supplemented with a human milk-based fortifier (445.93 ± 44.53 ug/mL) compared to unfortified maternal milk (301.8 ± 24.77 ug/mL; p=0.0379), which was not observed in maternal milk fortified with a bovine milk-based fortifier (304.75 ± 32.37 ug/mL; p>0.9999; Fig. 3C). Furthermore, when the fortifiers alone were evaluated for human IgA, human milk-based fortifiers contained significantly more IgA than the bovine milk-based fortifier, which contained negligible human IgA (p=0.0035; Fig. 3D).

## IgA retained in human milk-based fortifiers is cross-reactive to potential pathogens

To determine if IgA present in the human milk-based fortifiers retained functional activity, we tested the ability of IgA to react and bind common pathogenic bacteria *Streptococcus agalactiae* (also known as Group B *Streptococcus*, or GBS) and *Escherichia coli* by ELISA. We detected significant binding of IgA from human milk-based fortifiers to both GBS (p=0.0036; Fig. 4A) and *E. coli* (p=0.0036; Fig. 4B). Finally, we assessed binding of IgA from fortified and unfortified maternal milk to GBS and *E. coli*. While we did not observe a statistically significant increase in absorbance from the human milk-based fortified milk compared to unfortified milk with either GBS (p=0.2080; Fig. 4C) or *E. coli* (p=0.8315; Fig. 4D), milk with a human milk-based fortifier contained significantly more GBS-reactive IgA than milk with a

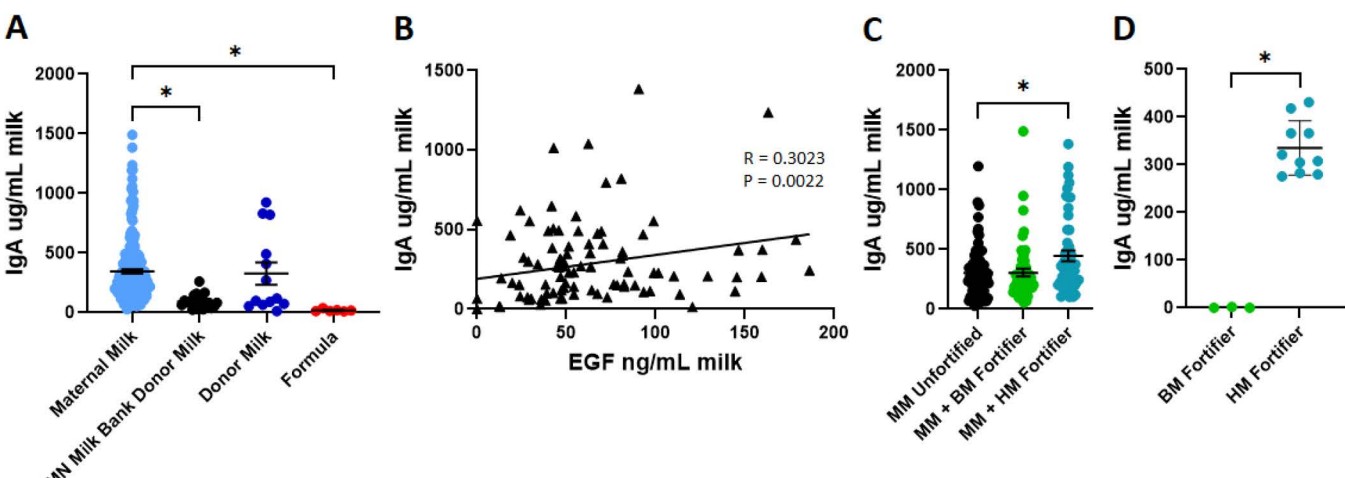

**Fig 3. IgA is decreased in individual donor milk and is present in human milk-based fortifiers.** A) Concentration of IgA in maternal milk (light blue, n=216), individual donor milk specimens (black, n = 20), pooled donor milk specimens (dark blue, n=13), and formula (red, n=6). B) IgA in milk plotted against EGF in milk from same specimen. C) Concentration of IgA in unfortified maternal milk (black, n=75), maternal milk fortified with bovine milk-based fortifier (green, n=55), and maternal milk fortified with human milk-based fortifier (blue, n=52). D) Concentration of IgA in bovine milk-based fortifier (green) and human milk-based fortifier (blue). * denotes significance, p<0.05, Kruskal-Wallis test with Dunn's multiple comparisons test in A and C, Pearson's correlation in B, and Mann-Whitney in D.

bovine milk-based fortifier (p=0.0026). We did observe a general trend of reduced IgA binding in the bovine milk-based fortified milk samples compared to the other two groups, though the statistical difference was only significant for GBS (p = 0.0117), and not *E. coli* (p=0.3030).

## Discussion

Our findings demonstrate that EGF and IgA content delivered to infants in the NICU can vary substantially depending on the diet type they receive and on further dietary additions like human milk fortifiers. Concerning diet choice, our results indicate a pooled donor milk diet may be more likely to provide greater EGF and IgA concentrations than individual donor milk or formula and would more closely reflect the concentrations of these factors delivered by a maternal milk diet. In addition, we found that human milk-based fortifiers retained significant levels of these biomolecules and contributed to an increase in EGF and IgA when added to milk. Together, these findings further underline the importance of dietary choices in early life as a source of biologically functional human milk factors.

When we analyzed donor milk specimens from individual donors, EGF and IgA were both significantly reduced compared to maternal milk specimens, while pooled milk samples were similar to maternal milk. This is consistent with previous observations reporting that single donor pools contained significantly less IgA than multi-donor milk pools [29]. Interestingly, we observed in our dataset that IgA concentrations in pooled donor milk mirrored those in maternal milk despite undergoing Holder Pasteurization, which has been shown to dramatically reduce IgA in treated milk [30], possibly as a result of the relatively small amount of pooled donor milk samples analyzed. One possible explanation for the difference in average EGF and IgA concentration between pooled donor milk and the individual donor milk samples that compose a pooled milk sample could be the presence of a small amount of high expressors in the donor pool. Milk component concentrations are highly variable between individuals [31] and can further vary based on prematurity [32], size for gestational age [33], and postpartum milk age [34] — though this may be a negligible factor for IgA specifically [29]. Together, these contributions could lead to some donor milk samples being much

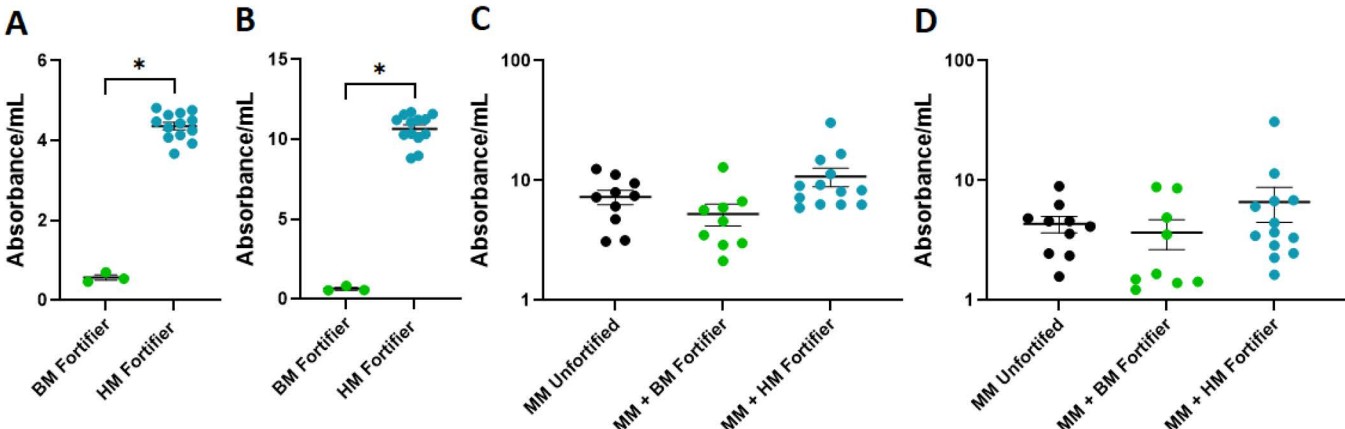

**Fig 4. IgA retained in human milk-based fortifiers is cross-reactive to potential pathogens.** A) Reactivity of IgA in bovine milk-based fortifier (green) and human milk-based fortifier (blue) to *Streptococcus agalactiae* (GBS). B) Reactivity of IgA in bovine milk-based fortifier (green) and human milk-based fortifier (blue) to *Escherichia coli*. C) Reactivity of IgA in unfortified maternal milk (black), maternal milk fortified with bovine milk-based fortifier (green), and maternal milk fortified with human milk-based fortifier (blue) to GBS. D) Reactivity of IgA in unfortified maternal milk (black), maternal milk fortified with bovine milk-based fortifier (green), and maternal milk fortified with human milk-based fortifier (blue) to *E. coli*. * denotes significance, p<0.05, Mann-Whitney test in A and B, and Kruskal-Wallis test with Dunn's multiple comparisons test in C and D.

more highly concentrated in certain milk factors than others, depending on the individual donor. The inclusion of one of these enriched samples in a donor milk pool would significantly increase the concentration of some milk factors in the final pooled sample, potentially bringing it more in line with average maternal milk. Our observed correlation between EGF and IgA concentrations in a given milk sample suggests that inclusion of one donor's milk that is highly concentrated in one milk factor may also result in increased concentrations of other beneficial factors in the final pooled donor milk. This could be a reflection of when donor milk was donated during the lactation cycle, which can be difficult to track when pooling individuals donor milk. However, a diet of individual donor milk still delivers more of these crucial biomolecules than a formula diet, which contains virtually no EGF or IgA. The capacity of pooled donor milk in replicating the EGF and IgA concentrations of maternal milk demonstrates that donor milk pooling could provide an immunological value that formula generally cannot, though larger cohorts of pooled donor milk-fed and formula-fed infants are needed to validate this trend.

While prior reports have measured key nutritional and immunological components from human milk at the time to expression and collection to understand human variation, we sought to measure EGF and IgA at the time of consumption by the infants. We consider measuring these components from prepared milk a strength of the study, as we were able to observe how fortifier use affected EGF and IgA concentrations. In addition to the macronutrients and minerals that human milk fortifiers aim to supplement, we observed human milk-based fortifiers contain meaningful amounts of human EGF and IgA. Human milk-based fortifiers, unlike the bovine milk-based fortifier we analyzed, preserved enough of both biomolecules that fortified milk was significantly enriched for EGF and IgA compared to unfortified maternal milk. Previous work has observed differences in milk factor concentrations between milk fortified with either human or bovine-derived fortifiers [35], though EGF has not been analyzed in this context and effects on IgA are inconclusive [36,37].

We also found that the human IgA present in the human milk-based fortifiers was able to bind to multiple pathogens implicated in neonatal sepsis, suggesting that IgA present in human milk-derived fortifiers could be protective against sepsis events resulting from translocation of pathogens in the infant gut [38]. This IgA may also play a role in shaping the composition of the developing microbiome [39], though further analysis and understanding of phenotypic differences associated with different dietary options is needed [40,41]. The impact of fortifier source has not clearly been established [42–45] and the recommendations for milk fortification for preterm infants in general are still being determined [46–48]. A clinical implication of our work is that the retention of beneficial components like EGF and IgA in human milk-based fortifiers could be considered for a diet that aims to retain human milk-derived immunological protection.

## Limitations

One limitation of this study is a lack of association between dietary options and health outcomes. Our study was unable to directly judge the health impacts of the various diets given its observational nature, as well as the rarity of neonatal infection outcomes. While our data concurs with other work reinforcing the value of pooling donor milk as a method to reduce individual variance between donor milk samples [29,31,34], further research is necessary to assess how the retention of critical factors in donor milk and fortifiers may contribute to the intestinal health of the infant. We also had significantly fewer donor milk samples than maternal milk samples in our dataset and were therefore unable to assess the contribution of different types of fortification to milk factor concentrations in donor milk. Due to the high

rate of maternal-milk fed participants our dataset is unable to assess how changing in diet, either from maternal milk or to maternal milk, may affect the stool concentration of these components. Future work will assess how these components may fluctuate throughout the first weeks of early life. A further limitation was our inability to ask if reduced EGF in individual donor milk was reflected in the stool of corresponding infants, as this study did not have enrolled neonates on an individual donor milk diet due to current nutritional practices at the study site's location. We did, however, find a significant amount of EGF in the stool of infants fed human milk (maternal or pooled donor) as compared to formula, which suggests that infants ingesting greater levels of EGF are also passing that EGF through their digestive tract. How this milk EGF is utilized biologically by the neonate through the gastrointestinal tract as a growth factor remains an active area of interest. One possible option to further investigate the functional significance of increased EGF and IgA would be to analyze the stool microbiome compositions of our samples and relate the results to the amount of EGF and IgA in each infant's diet. Finally, as our primary focus was infants' exposure to milk factors from their prepared diets, we did not account for potential influences from processing factors upstream of diet delivery.

## Acknowledgments

All authors critically reviewed and approved the manuscript.

## Author contributions

**Conceptualization:** Emily Levy, Jane E. Brumbaugh, Kathryn Knoop.

**Data curation:** Kathryn Knoop.

**Formal analysis:** Kara Greenfield.

**Funding acquisition:** Kathryn Knoop.

**Investigation:** Christian Tamar, Kara Greenfield, Katya McDonald, Kathryn Knoop.

**Methodology:** Christian Tamar, Katya McDonald, Emily Levy, Kathryn Knoop.

**Project administration:** Jane E. Brumbaugh, Kathryn Knoop.

**Resources:** Emily Levy, Jane E. Brumbaugh.

**Supervision:** Kathryn Knoop.

**Validation:** Kara Greenfield.

**Visualization:** Christian Tamar, Kara Greenfield, Kathryn Knoop.

**Writing – original draft:** Christian Tamar, Kathryn Knoop.

**Writing – review & editing:** Christian Tamar, Emily Levy, Kathryn Knoop.

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
