## [Decision Letter · Decision Letter 0]

16 Dec 2024

PONE-D-24-48127EGF and IgA in maternal milk, donor milk and milk fortifiers in the Neonatal Intensive Care Unit settingPLOS ONE

Dear Dr. Knoop,

Thank you for submitting your manuscript to PLOS ONE. After careful consideration, we feel that it has merit but does not fully meet PLOS ONE’s publication criteria as it currently stands. Therefore, we invite you to submit a revised version of the manuscript that addresses the points raised during the review process.

**ACADEMIC EDITOR: **

The manuscript offers valuable insights into EGF and IgA levels in maternal and donor milk alternatives for premature infants and is well-structured overall. However, several concerns need to be addressed to enhance the clarity and strength of the paper:

1) Justification for Infant Participant Enrollment: Please explain why infants were enrolled starting at 3 days post-birth and why sample collection began at 8 days. This delay may impact EGF and IgA concentrations, so a brief justification (2-3 sentences) supported by evidence (if available) would be helpful.

2) Statistical Analysis: Provide a clear rationale for the statistical tests used, including how the error rate for multiple comparisons was managed, to ensure the robustness of the findings.

3) Sample Size: A detailed explanation of how the sample size was determined would help clarify the study's statistical power and relevance.

4) Ethical Statement on Donor Milk: Please include a statement addressing the ethical considerations surrounding the use of donor milk from the milk bank, ensuring transparency and adherence to ethical standards.

5) Strengths and Clinical Implications: Clearly outline the strengths of the study and discuss its clinical implications, emphasizing how the findings could impact neonatal care practices and improve outcomes for premature infants.

We look forward to receiving your revised manuscript.

Kind regards,

Anh Nguyen

Academic Editor

PLOS ONE

2. Thank you for stating the following financial disclosure: [KK NIH DK134366]. At this time, please address the following queries:

3. We note that your Data Availability Statement is currently as follows: [All relevant data are within the manuscript and supporting files.] Please confirm at this time whether or not your submission contains all raw data required to replicate the results of your study. Authors must share the “minimal data set” for their submission. PLOS defines the minimal data set to consist of the data required to replicate all study findings reported in the article, as well as related metadata and methods (https://journals.plos.org/plosone/s/data-availability#loc-minimal-data-set-definition).

Additional Editor Comments:

The manuscript offers valuable insights into EGF and IgA levels in maternal and donor milk alternatives for premature infants. However, several concerns need to be addressed to enhance the clarity and strength of the paper:

1) Justification for Infant Participant Enrollment: Please explain why infants were enrolled starting at 3 days post-birth and why sample collection began at 8 days. This delay may impact EGF and IgA concentrations, so a brief justification (2-3 sentences) supported by evidence (if available) would be helpful.

2)Statistical Analysis: Provide a clear rationale for the statistical tests used, including how the error rate for multiple comparisons was managed, to ensure the robustness of the findings.

3) Sample Size: A detailed explanation of how the sample size was determined would help clarify the study's statistical power and relevance.

4) Ethical Statement on Donor Milk: Please include a statement addressing the ethical considerations surrounding the use of donor milk from milk banks, ensuring transparency and adherence to ethical standards.

5) Strengths and Clinical Implications: Clearly outline the strengths of the study and discuss its clinical implications, emphasizing how the findings could impact neonatal care practices and improve outcomes for premature infants.

Reviewers' comments:

Reviewer's Responses to Questions

**Comments to the Author**

1. Is the manuscript technically sound, and do the data support the conclusions?

Reviewer #1: Partly

Reviewer #2: Yes

2. Has the statistical analysis been performed appropriately and rigorously? 

Reviewer #1: I Don't Know

Reviewer #2: Yes

3. Have the authors made all data underlying the findings in their manuscript fully available?

Reviewer #1: Yes

Reviewer #2: Yes

4. Is the manuscript presented in an intelligible fashion and written in standard English?

Reviewer #1: Yes

Reviewer #2: Yes

5. Review Comments to the Author

Reviewer #1: This is a prospective cohort study comparing EGF and IgA concentrations as important infant health markers, in various sources of milk for premature infants.

The aim behind the study is clear and the significance of the study outcomes may help contribute towards decision making in NICU around premature infant nutrition. However, there are a number of missing information and methodological vagueness that needs to be addressed:

- One of the main objectives of the study was to analyze milk fortifiers. However, very little information is provided about what these are and how they are used in NICU. More information needs to be provided in the introduction and method section

o The concentration of EGF in the HM fortifier is around 100ng/ml and yet when it was added to the maternal milk, it only increased EGF by 20ng/ml. What was the dilution factor for maternal milk to fortifier? This needs to be described in the methods section. And can’t one simply add more of the fortifier if needed?

o Other information that needs to be provided includes what is the shelf life of fortifiers? How easy is it to get them and is there realistically enough to be used in every NICU?

- Have the authors recorded the age from parturition of the individual milk donated? This may explain the difference between the pooled and individual milk concentration of EGF and IGA. If it hasn’t been recorded, an explanation to why that is needs to be provided

- Which infant formulas have been used? There are many kinds and at the very least there needs to be a discussion about how the infant formula used in this NICU may differ in formulation from other formulas out there

- There were 74 participants and 237 samples. Does that mean that an individual infant would have had different diets at different times? For example, can one infant be given maternal milk one day and formula the other day? If so, has this been recorded, and could it have affected the results from the stool samples?

- In the first section, maternal milk has 59.72 ng/ml EGF, in the other section it is 49.11 ng/ml EGF. Why is it different? Did the authors measure before and after fortification in the second section? If so, this needs to be described clearly in the methods section. Similarly for IGA.

- More information needs to be provided about how each diet is received, where it is sourced from and prepared for the infant.

- For pearson correlations, an R=0.1885 is not significant.

Other points:

- P-values should be provided in the text of the results section and abstract.

- The methodology section needs re-writing to be more organized and detailed.

- Is there a reason why stool was not analyzed for IGA?

- “Our observed correlation between EGF and IgA concentrations in a given milk sample suggests that inclusion of individual donor milk highly concentrated in one milk fact within a donor pool may increase the concentration of other beneficial factors in the final pooled donor milk.”---Not clear what this means

- Translocation to where?

- In the introduction, authors mentioned a pre-clinical model regarding milk expressed closer to parturition. This needs to be expanded on the type of model used.

- Need to define what “diet” means for the purpose of the study. Since both mum and infant were recruited in the study. Similarly, define what “participant” refers to in the study.

- “Clinical data was de-identified”. When was it de-identified and who was blinded to this information?

- “Incomplete sample pairs were discarded”. More information needs to be provided here. What does that mean? Who made the decision to discard?

- The small para about IgA bacterial binding in the methods sections, needs to be expanded and more information about the method provided clearly.

- How was the poop samples homogenized?

- The article could do with an editorial review to correct grammatical errors throughout.

Reviewer #2: The paper by Tamar et al. contributes a survey on the content of EGF and IgA in maternal milk alternatives, like donor human milk or formula, which supports the value of providing donor human milk to infants born pre-term or low birth weight. The findings of the work underline the importance of dietary choice in early life as a source of biologically functional human milk factors. Another important observation of the study is the significant increase of EGF and IgA concentrations in pooled donor milk as compared to individual donor milk samples. Moreover, the correlation between EGF and IgA concentrations in the various milk sample suggests that individual donor milk highly concentrated in one milk factor may also contain increased concentrations of other beneficial factors. Human milk-based fortifiers, unlike the bovine milk-based fortifiers, preserved significantly enriched EGF and IgA levels as compared to unfortified maternal milk. Finally, the authors show that IgA present in human milk-derived fortifiers binds common pathogenic bacteria and could therefore be protective against neonatal sepsis. The data are consistent with other works and support the notion that human milk-derived immunological protection could be fostered by beneficial components like EGF and IgA contained in human milk-based fortifiers. In addition, the paper underscores the value of pooling donor milk for securing optimal levels of beneficial biomolecules to feed the infant. The data are clearly presented and accurate; they constitute a useful resource for managing infants in NICUs.

6. PLOS authors have the option to publish the peer review history of their article (what does this mean? ). If published, this will include your full peer review and any attached files.

**Do you want your identity to be public for this peer review?** For information about this choice, including consent withdrawal, please see our Privacy Policy .

Reviewer #1: No

Reviewer #2: No

---

## [Author Response · Author response to Decision Letter 0]

10 Feb 2025

We thank the reviews and editors for their time and effort in the review of our work. Below is a point-by-point response, italicized, to each comment with our action taken in the revised manuscript.

ACADEMIC EDITOR:

1) Justification for Infant Participant Enrollment: Please explain why infants were enrolled starting at 3 days post-birth and why sample collection began at 8 days. This delay may impact EGF and IgA concentrations, so a brief justification (2-3 sentences) supported by evidence (if available) would be helpful.

We agree that EGF and IgA, along with other components in the breast milk fluctuate significantly in the initial days of lactations as colostrum transitions to mature milk. For the point of this study, we have included the justification “sample collection began at 8 days, avoiding colostrum sampling which is known to have different composition compared to transitional and mature milk.” (Line 102), and cited supporting literature.

2) Statistical Analysis: Provide a clear rationale for the statistical tests used, including how the error rate for multiple comparisons was managed, to ensure the robustness of the findings.

We have expanded the section on statistical tests used in this study in the methods section (line 166), and confirmed these tests are also listed in the figure legends of the corresponding figure panels.

3) Sample Size: A detailed explanation of how the sample size was determined would help clarify the study's statistical power and relevance.

This information has been added to the human subjects section (Line 103-105). In short, sample size reflects all participants consented during a two year period for data analysis of samples collected during that time. As this was a prospective observation study, all mother-infants pairs consented were included regardless of infant diet during the duration of the study. This resulted in unequal numbers in the various diet groups, and has been listed as a limitation of this study (line 351).

4) Ethical Statement on Donor Milk: Please include a statement addressing the ethical considerations surrounding the use of donor milk from the milk bank, ensuring transparency and adherence to ethical standards.

Thank you for this recommendation. The OhioHealth Mothers Milk Bank is a member of the Human Milk Banking Association of North America, which “promotes collection and distribution of donor human milk in a safe, ethical and cost effective manner”. We have added an ethical statement on the use of donor milk (line 134).

5) Strengths and Clinical Implications: Clearly outline the strengths of the study and discuss its clinical implications, emphasizing how the findings could impact neonatal care practices and improve outcomes for premature infants.

We have included these points in the discussion (line 322, and line 339).

Reviewer #1: This is a prospective cohort study comparing EGF and IgA concentrations as important infant health markers, in various sources of milk for premature infants.

The aim behind the study is clear and the significance of the study outcomes may help contribute towards decision making in NICU around premature infant nutrition. However, there are a number of missing information and methodological vagueness that needs to be addressed:

- The article could do with an editorial review to correct grammatical errors throughout.

Response: We thank the reviewer for their time in reviewing our work. We have revised the manuscript throughout to correct grammatical errors.

- In the first section, maternal milk has 59.72 ng/ml EGF, in the other section it is 49.11 ng/ml EGF. Why is it different? Did the authors measure before and after fortification in the second section? If so, this needs to be described clearly in the methods section. Similarly for IGA.

Response: In our first results section pertaining to figure 1, the maternal milk EGF value reported reflects all maternal milk, both fortified and unfortified. In the second section, the 49.11 ng/mg value reported for EGF is for unfortified maternal milk only. This is the same for IgA measurements. We have revised the figures to make this clear.

- “Our observed correlation between EGF and IgA concentrations in a given milk sample suggests that inclusion of individual donor milk highly concentrated in one milk fact within a donor pool may increase the concentration of other beneficial factors in the final pooled donor milk.”---Not clear what this means

Response: We have clarified this sentence and expanded our discussion to incorporate comments regarding age from parturition of the individual milk donated (line 313) as the reviewer points out below in a separate comment.

- Translocation to where?

Response: We have clarified this concept to discuss how bacteria can translocate from the lumen through the intestinal epithelium and gain access to the bloodstream in the introduction (Line 46-47).

- In the introduction, authors mentioned a pre-clinical model regarding milk expressed closer to parturition. This needs to be expanded on the type of model used.

Response: We agree that this is an important clarification that needs to be made, as mammalian lactation can differ greatly between individual species. We have clarified in the introduction what observations are from mouse models, and what observations are from human data (Line 61).

- Need to define what “diet” means for the purpose of the study. Since both mum and infant were recruited in the study. Similarly, define what “participant” refers to in the study.

Response: We have defined infant diets in “Milk and Stool Collection” section of Methods, and included a definitional sentence (Line 121) to clarify we are measuring the prepared milk or formula provided to the infants. Diet of the mothers was not measured in this study. We have also defined participants refer to infants, though consent was obtained from the parents (line 91, section Human Subjects).

- One of the main objectives of the study was to analyze milk fortifiers. However, very little information is provided about what these are and how they are used in NICU. More information needs to be provided in the introduction and method section

Response: Thank you for this comment, we have included more information in the introduction in order to inform the reader of how these fortifiers are used in the NICU setting (beginning at line 72).

- The concentration of EGF in the HM fortifier is around 100ng/ml and yet when it was added to the maternal milk, it only increased EGF by 20ng/ml. What was the dilution factor for maternal milk to fortifier? This needs to be described in the methods section. And can’t one simply add more of the fortifier if needed?

Response: We have included more information regarding how fortifiers are added to milk in the methods section “Fortifiers and Formula” (Line 107). According to the manufacturers labeling, fortifiers are designed to add to human milk as specified dilutions, and improper dilution may be harmful to infants. We have included this important information.

- The small para about IgA bacterial binding in the methods sections, needs to be expanded and more information about the method provided clearly.

Response: We have expanded the information provided regarding our bacterial-binding ELISA in the methods for transparency (Line 160-165).

- How was the poop samples homogenized?

Response: Thank you for this reminder, we have included information in the methods section describing the stool homogenization. (Line 146).

- P-values should be provided in the text of the results section and abstract.

Response: Thank you, we have provided key p-values in the text.

- Other information that needs to be provided includes what is the shelf life of fortifiers? How easy is it to get them and is there realistically enough to be used in every NICU?

Response: Thank you for the opportunity to inform the readers more regarding NICU, we have incorporated how these are predominantly used, shelf life, and current usage among NICUs in the introduction (Life 77-80).

- Have the authors recorded the age from parturition of the individual milk donated? This may explain the difference between the pooled and individual milk concentration of EGF and IGA. If it hasn’t been recorded, an explanation to why that is needs to be provided

Response: The date the individual milk was expressed during lactation in reference to partition, was recorded, and range throughout lactation. We agree that donor milk is often donated later in lactation, and could reflect the differences in components. We have revised the discussion to clarify this point (Line 313). Due to the way donor milk is provided to the clinic, information regarding donors is collected by the donor milk bank, in this case, the OhioHealth Mother’s Milk Bank. Milk is pooled at the milk bank, and then provided to practices, including the Mayo Clinic NICU facilities.

- Which infant formulas have been used? There are many kinds and at the very least there needs to be a discussion about how the infant formula used in this NICU may differ in formulation from other formulas out there

Response: The few infant participants provided infant formula, and were given either Enfamil Enfacare 22 calories per ounce, Similac Specialcare 24 calories per ounce. While compositions differ between formulas, to our knowledge, no infant formula contains human EGF or supplement Ig molecules. We have included this information in the methods, along with the statement, “Neither of these formulations have added epidermal growth factor or immunoglobulins, and therefore are not expected to contain human EGF or IgA.” (Line 119-120)

- There were 74 participants and 237 samples. Does that mean that an individual infant would have had different diets at different times? For example, can one infant be given maternal milk one day and formula the other day? If so, has this been recorded, and could it have affected the results from the stool samples?

Response: The 237 samples from the 74 participants represent different temporal samples throughout the participants stay within the NICU. Each participant has at least two samples from different weeks, with an average 3 weekly milk-stool samples per participant. Diets from individual weeks have been recorded, shifts of diets between rarely occurred in individual participants, and no participant in this study moved from an exclusive maternal milk diet to an alternative donor milk or formula diet. We have included this topic as a discussion point (line 351-354).

- More information needs to be provided about how each diet is received, where it is sourced from and prepared for the infant.

Response: We have expanded the milk and diet information in the methods. In brief, all infants in this study received enteral feeds by bottle. Maternal milk is sourced from infants mother. Donor milk supplied to infants was sourced from the OhioHealth Mothers Milk Bank. Fortifiers were added per clinical care team recommendations prior to feeding. Diet was prepared by the nutrition team at the Mayo Clinic. Samples were obtained following fortification (if applicable) from prepared milk. This has been included in the “Fortifiers and Formula” section (line 107) and the “Milk and Stool Collection” section in the methods (Line 121).

- For Pearson correlations, an R=0.1885 is not significant.

Response: We agree that the Pearson’s correlation where R=0.1885 describes a weak positive linear correlation, and that the p-value of this correlation is p=0.0208 was significant. We have clarified the text to remove any implication of a stronger R (line 195).

- The methodology section needs re-writing to be more organized and detailed.

Response: We have expanded the methods section based on reviewers and editors comments. This section now includes more detailed information and has been reordered.

- Is there a reason why stool was not analyzed for IGA?

Response: We agree that the amount of IgA found in stool and how it corresponds to the diet is of interest. IgA often binds bacteria in the intestinal tract of the infant, therefore a significant amount of Ig in the intestine will be bound and not free. Measurement of free Ig in the stool could be combined with measuring of IgA bound to bacteria to understand dynamics of how milk-derived IgA can shape the developing microbiota. These studies will be the focus of future work.

- “Clinical data was de-identified”. When was it de-identified and who was blinded to this information?

Response: Clinical data was de-identified at the time of consent by the study coordinator, and research technicians performing assays were blinded to information that would allow for the identification of participants. This has been added to the human subject section (line 100-101).

- “Incomplete sample pairs were discarded”. More information needs to be provided here. What does that mean? Who made the decision to discard?

Response: Stool or Milk without corresponding stool or milk from the same week was not included in this analysis. Decision was made by the Principal Investigator as per the study design. This information has been added to the methods (Line 127-130).

---

## [Editor Report · Decision Letter 1]

14 Mar 2025

EGF and IgA in maternal milk, donor milk, and milk fortifiers in the Neonatal Intensive Care Unit setting

PONE-D-24-48127R1

Dear Dr. Knoop,

We’re pleased to inform you that your manuscript has been judged scientifically suitable for publication and will be formally accepted for publication once it meets all outstanding technical requirements.

Kind regards,

Anh Nguyen

Academic Editor

PLOS ONE

Additional Editor Comments (optional):

We appreciate your revised submission and your patience during the review process. The manuscript is interesting, and you have satisfactorily addressed the comments. We are pleased to accept it for publication.
---

## [Editor Report · Acceptance letter]

PONE-D-24-48127R1

PLOS ONE

Dear Dr. Knoop,

I'm pleased to inform you that your manuscript has been deemed suitable for publication in PLOS ONE. Congratulations! Your manuscript is now being handed over to our production team.

Kind regards,

on behalf of

Dr. Anh Nguyen

Academic Editor

PLOS ONE